# Relevant Factors in the Schooling of Children with Autism Spectrum Disorder in Early Childhood Education

**DOI:** 10.3390/brainsci14121167

**Published:** 2024-11-22

**Authors:** Francisco Villegas Lirola, Antonio Codina Sánchez

**Affiliations:** Department or Education, University of Almeria, 04120 Almeria, Spain; acodina@ual.es

**Keywords:** autism spectrum disorder, schooling modality, difficulties in developing, preschool education

## Abstract

Background: Educational professionals face significant challenges in determining the most appropriate educational placement for each child with ASD, which is a major concern for their parents. The purpose of this paper is to identify the factors in the development of students with ASD (language development, cognitive development, and socio-emotional development) that are most relevant in determining the modalities of schooling in early childhood education. Methods: A total of 381 Psychopedagogical Evaluation Reports from students with ASD aged 3 to 5 years were reviewed. The importance of each variable was identified using artificial neural network analysis. Classification trees were used to determine their distribution in the two schooling modalities. Results: A total of 42.9% of boys aged 3–5 years with ASD and 54.12% of girls aged 3–5 years with ASD were enrolled in specific modalities. Conclusions: Cognitive development and comprehensive language were the two variables that best explained whether children with ASD were educated in specific or ordinary modalities. The presence of a significantly impaired level of cognitive development was the best predictor of schooling in specific modalities, and for the rest of the cognitive levels, the greater the difficulties in comprehensive language, the greater the likelihood of schooling in specific modalities.

## 1. Introduction

The term “autism spectrum disorders” (ASD) encompasses a continuum of disorders with different biological origins that affect the ability to initiate and maintain reciprocal interaction and social communication. They are characterized by restricted, repetitive, or inflexible patterns of behavior and interests [1]. The presence of additional neurodevelopmental conditions, including mental retardation, language impairment, and the severity of the impairment in relation to the need for assistance, should be indicated.

As outlined in the DSM-5-TR, the prevalence of this condition varies from 1 to 2% in the United States and approximately 1% in other countries [2]. In 2020, the Autism and Developmental Disabilities Monitoring (ADDM) estimated a prevalence rate of 27.6‰ in the United States [3]. For the period between 2021 and 2022, Yan et al. [4] reported a prevalence of 37.9‰ (one in 26) in the United States.

In Spain, in 2017, Fuentes et al. [5] found a prevalence of 5.9‰ in the province of Guipuzcoa. In Catalonia, Pérez-Grespo et al. [6] estimated a prevalence of 12.3‰ for 2017, while Bosch et al. [7], also for Catalonia, found a prevalence of 7.0‰. Additionally, Morales-Hidalgo et al. [8] reported 7.1‰ in Tarragona (Catalonia) for 2017.

In the context of the research conducted in the province of Almería, Spain, Ville-Gas-Lirola [9] reported a prevalence of 8.65‰ in Almería (Andalusia) for the year 2020. For the year 2021, the average prevalence across all provinces of Andalusia was 8.10‰ [10]. In 2024, Villegas-Lirola and Codina-Sánchez [11] observed a prevalence of 13.68‰ for Almería (Spain). It is anticipated that the number of children with ASD will continue to increase.

The transition to school represents a significant challenge for families and necessitates considerable effort from educational administrators. The difficulty in reaching the various milestones of the child’s neurodevelopment requires the provision of specific interventions and resources in the schooling of children with ASD. As children with ASD begin their schooling, it is necessary to know how their psycho-evolutionary characteristics are relevant to the schooling modality proposed to them.

### 1.1. Language y ASD

The language skills of children with ASD exhibit considerable variability, which can be attributed to the high degree of heterogeneity observed in this population. This heterogeneity encompasses a range of abilities, from non-verbal communication to the ability to express oneself effectively in verbal situations [12].

In this sense, approximately 30% of school-age children (three years old) with ASD are minimally verbal [13]. It is also noteworthy that at two years of age, 14.7% have not acquired any words [14], and that 87% of children with ASD have language delay at age three [15]. In addition, only 16% of children with ASD between 4 and 18 years have no language impairment [16].

Children’s language interactions with adults, as well as between children themselves, in which they exchange words and sounds in an organized way, facilitate the creation of shared meanings, which are fundamental to the development of language [17,18]. In contrast, behaviors such as simultaneous speech limit the development of language skills (perception, attention, integral and expressive language), thereby reducing the potential for learning opportunities [19,20]. Furthermore, the acquisition of knowledge is not only delayed but also atypical [21]. In addition, vocal exchange skills are often affected in children with ASD [19], and the ability to speak pragmatic language is also affected, irrespective of their language development level and age [22].

### 1.2. Socio-Emotional Development and ASD

The presence of difficulties in social interaction and communication is central to the diagnosis of ASD [2] and is often evident from an early age [23]: from the perspective of social interaction and communication, social communication difficulties have a negative impact on the behavior of children in relation to their peers [24]. Children with ASD frequently encounter difficulties in differentiating emotions, regulating affect, and forming relationships with other children [25].

Moreover, one of the primary sources of stress in the relationship between teachers and children with ASD, as well as in the relationship between families and schools, is typically the presence of disruptive behaviors [26], which may be associated with difficulties in the executive function of self-regulation [27].

### 1.3. Cognitive Development and ASD

Cognitive delay is a common phenomenon among individuals with autism spectrum disorder, and “even those with normal or high intelligence often have an uneven ability profile” [2] (p. 62). The prevalence of intellectual disability in individuals with ASD has been estimated to range from 33% to 70% [3,28].

The quantification of cognitive abilities is achieved through the administration of intelligence tests, which are based on the assumption that enhanced performance on said tests requires the operation of improved executive functions in the context of everyday life [29], as well as the capacity to adapt to new and changing circumstances [30].

The initial five-year period is of particular significance regarding the development of executive functions [31], which serve to establish the foundation for the subsequent maturation of high-level cognitive processes that will emerge during adolescence and adulthood [32].

Children with ASD frequently exhibit challenges in inhibitory functioning, working memory, and flexibility [33]. These challenges manifest as difficulties in the capacity to voluntarily inhibit or block irrelevant responses or information [34], remember sequential information [35,36], tolerate change, voluntarily switch attention, solve problems, and approach tasks in a flexible manner considering the context or particular circumstances [37]. Therefore, it can be concluded that executive functioning serves as a reliable indicator of future educational prospects for children diagnosed with autism spectrum disorder [38].

### 1.4. Schooling, Mental Health, and ASD

Deficits in ASD result in clinically significant impairment across a range of functional, social, academic, vocational, and daily living areas of an individual [1,2]. When parents of children with ASD engage with the educational system, they frequently encounter stressors [39].

At pivotal transition points, such as when the child is preparing to enter school (3 years old), after kindergarten (0–2 years old), parents of children with ASD experience heightened levels of uncertainty and anxiety, “Where do the autistic kids go in the borough? What is the best school for the autistic kids? But no one will talk to you” [40] (p. 1104). These are not easy decisions. Families experience the process with uncertainty and obstacles:

These are times of great tension because families have to become virtual experts in the technical and bureaucratic schooling procedures that they will have to go through, which are particularly complex in the case of students with autism spectrum disorder, given the existing schooling options[41] (p. 134).

The mental health needs of children with ASD are associated with their difficulties in the educational setting [42] (Heyman et al., 2020). While schooling is a significant concern for parents in general, for those with children who have been diagnosed with autism spectrum disorder, the experience is often considerably more complex and challenging. Research indicates that parents of children with ASD frequently report feelings of worry, stress, anxiety, and frustration when they send their children to school [43] (Stoner et al., 2015). These are the main reasons why the identification and intervention, both medical and educational, of children with ASD must be the result of a collaborative effort between health and education services.

It is imperative that professionals avoid any inconsistencies in their approach. When there are discrepancies between the advice provided by health and educational professionals, it can lead to confusion and a lack of trust in the decisions made by health or educational psychiatrists and psychologists regarding their children’s care [44] (Tucker and Schwartz, 2013).

In our case, this coordination between health and education actions was achieved through a collaborative effort between the Child and Adolescent Mental Health Unit and the Educational Guidance Team, which specialized in autism spectrum disorders, in conjunction with the Coordinator of the Special Educational Needs Area of the Provincial Technical Team for Educational Guidance.

This collaboration allows all children with ASD who are initially identified by the Child and Adolescent Mental Health Unit to receive an educational diagnosis. Furthermore, it allows children with ASD who are first identified in an educational setting to undergo a diagnostic evaluation from the Child and Adolescent Mental Health Unit.

In Spain, the schooling of children with special educational needs (SEN) is subject to a standardized procedure across all autonomous communities. This procedure requires the assessment of appropriate counseling services, resulting in the completion of two documents: a psychopedagogical evaluation and a schooling report.

A psychopedagogical evaluation is a collaborative process conducted by the Multidisciplinary Guidance Team of the Educational Centre. It involves teachers, students, the family, and, where appropriate, professionals from other health and social services. The purpose of this process is to integrate all useful information to support decisions about the selection of educational interventions and resources [45]. The results are included in the psychopedagogical evaluation report, which is a standardized document that specifies the educational needs, the developmental and educational situation of the student in interaction with the developmental and educational contexts [46], as well as the measures, resources, and educational support that may be needed during their schooling.

The Schooling Report is a technical, well-founded, and synthetic report of psychopedagogical evaluation. It includes the proposed type of schooling, assistance, educational support, and specific human, material, or technical resources required. It is important to gather the family’s opinion, regardless of whether they agree with the content of the school report, particularly with regard to the type of schooling proposed [47]. In cases where regular schooling is not feasible, it is essential to provide a rationale for this decision and to justify the choice of a special education center as an alternative [48].

### 1.5. Special Education Student Enrollment Procedures

During the regular school period (from March to the end of April), families of children with SEN submit applications for school places. Educational guidance services conduct assessments of the children’s educational requirements. Families then apply to the educational centers that they prefer and that have the resources that their children require. If the educational commissions of the zone are unable to reach a decision on these proposals, they will be forwarded to a provincial commission for resolution (Figure 1).

In the event of exceptional educational requirements (beyond the typical school schedule), the Provincial Service formulates a proposal based on the availability of enrollment and the necessary resources at the centers in the student’s area of residence, as well as the preferences of their family.

In the Community of Madrid, this work is undertaken by the Specific Schooling Support Service [49]. In the Community of Valencia, a Personalized Action Plan (PAP) is formulated based on the findings of the psychopedagogical evaluation, with the objective of organizing, developing, and evaluating an educational proposal for the inclusion of students with SEN.

### 1.6. Psychopedagogical Evaluation Report

In order to ascertain the specific educational requirements of a student and the resources that may be necessary to facilitate their learning, a psychopedagogical evaluation report is prepared. In addition to information regarding the fulfillment of the diagnostic criteria for ASD, the Psychopedagogical Evaluation Report includes information pertaining to the child’s clinical and educational history, as well as the student’s personal development. This encompasses cognitive development, psychomotor development, personal autonomy skills, communicative intentionality, expressive language, comprehensive language, and social and emotional development.

The resources required for effective education, both personal and material, are included in the report, as well as guidance for families and educators. Additionally, a proposal for an optimal schooling modality is presented.

Furthermore, the Psychopedagogical Evaluation Report identifies the individual responsible for its preparation, the participants, the commencement and conclusion dates, and the date of information dissemination to the family. The opinion of the family is included in an appendix, as well as all documentation provided by the family and other professionals.

Schooling modalities.So, everybody, everybody always tells us that we should cry and kick and scream and everything because our son is going into a B mode, okay? usually with support, and I, well, obviously, given the relationship and the trajectory that our son has in kindergarten with the rest of his classmates, well, obviously what I would want for my son is a B mode, okay? That’s one thing. But on the other hand, (the private Early Care Center) has also told us that what they recommend for our son is a C modality, a TEA classroom. Well, what do you want me to say… that if these professionals are already telling us that, well… it’s not that I don’t want him to go to a TEA classroom, it’s that I’m saying, well damn, … if our son is really capable of being in a regular classroom with support… (M.G., 2 years old)[41] (p. 128).

In the event that a school place is requested for students with special educational needs, a report is prepared or updated in advance of the placement. A proposed schooling modality is presented as a measure of attention to exceptional or extraordinary diversity [50]. The most inclusive form of schooling is the ordinary school [51]. In instances where this is not feasible, and there are reasonable grounds for doing so, education may be provided in ordinary preferential care centers, special units, or special education centers [52].

With regard to students of childhood education age (3–6 years), it is standard practice for them to be educated in ordinary schools. However, in very exceptional cases, alternative forms of schooling may be employed [53].

The schooling of SEN students (Figure 2) can be provided in two different modalities. These include specific modality (special education center or specific unit within a regular center. The latter may take the form of a stable classroom, an enclaved classroom, a specialized or general open classroom), and ordinary modality (regular full-time or variable periods).

Specific schooling should only be considered for students who require a high-intensity educational response that is difficult to generalize. This approach necessitates the provision of highly individualized attention and the utilization of highly specialized resources.

Ordinary schooling modalities (full-time schooling and variable-time schooling) entail the enrollment of students in early childhood education, primary education, and compulsory secondary education. Upon the successful completion of the latter, students are awarded the qualification of a graduate of the compulsory secondary education stage.

In specific schooling modalities (schooling in a specific classroom or schooling in a specific center), students are enrolled in Specific Basic Training (Specific Early Childhood Education, Compulsory Basic Training, and Transfer to Adulthood and Working Life Program) and cannot complete their studies and obtain the qualification of Graduate in Compulsory Secondary Education. Failure to obtain the Secondary Education Graduate Title has adverse implications for their social and professional future.

### 1.7. Research Objectives

A review of the literature reveals a paucity of studies that permit the identification of the type of schooling in which students with ASD are typically enrolled based on the characteristics of their personal development (cognitive, linguistic, and socio-emotional) [54]. The issue of schooling at the age of three is a significant concern for families who experience the evaluation and schooling process for their children with ASD with suspicion and a sense of helplessness:

It is that (in the Psychopedagogical Evaluation Report) the future of your child is defined in a photo, which is what this woman (the counselor) sees at a time [...]. You get the first assessment of the inexpert, you believe everything, your critical ability is not because what they are telling you is so hard that you do not have time to think, Everything falls on you, you don’t know what to do and you go home without being able to react [PN-3 years][41] (p. 130).

The purpose of this paper is (1) to ascertain the personal factors influencing the cognitive, linguistic, and socio-emotional development of children with ASD in early childhood education and to determine the extent to which these factors determine the modality of their education. And in this case, (2) to identify the factors (age, gender, cognitive development, psychomotor development, communicative intention, expressive language, comprehensive language, and socio-emotional development) that have a greater influence on the chosen schooling modality.

## 2. Materials and Methods

### 2.1. Participants and Data Sources

In the province of Almeria (Spain), 1522 students with ASD (2024) have been identified by the educational administration. Of these, 386 are between the ages of 3 and 5. In order to identify the relevant factors involved in the schooling of students with ASD in early childhood education, the psychopedagogical evaluation reports (which have been anonymized by the educational services) of 381 students for whom informed consent was obtained (296 boys and 85 girls) were subjected to analysis (Figure 3).

This study was conducted in accordance with the ethical standards set forth in the Declaration of Helsinki and received approval from the Bioethics Committee of the University of Almería (UALBIO2022/058).

### 2.2. Procedures for Diagnosing and Enrolling Students with Autism Spectrum Disorder

Who performed the assessments? In Almería (Spain), the personnel of the Educational Guidance Teams (EGT) are the responsible parties for carrying out the diagnosis and registration of students with autism spectrum disorder in the Educational Administration. The Educational Guidance Teams are regional in nature, serving students in early childhood and primary education from a number of nearby communities (Figure 4).

The evaluations were conducted by the staff of the sixteen Educational Guidance Teams, comprising a total of 146 counselors. The identification and diagnosis were carried out in collaboration with the consultants of the Educational Guidance Team specializing in autism spectrum disorders between 2022 and 2023.

The assessments were conducted based on the identification of the cases by the Health Centers and Child and Adolescent Mental Health Units. In cases initially identified at educational centers, referrals were made to the relevant Child and Adolescent Mental Health Units.

The Educational Guidance Teams and the Specialized Educational Guidance Team for ASD are responsible for the diagnosis and registration of students with ASD in the Educational Administration. Psychologists and psychiatrists from the Zonal Mental Health Units, especially the Child and Adolescent Mental Health Unit, are responsible for diagnosing and registering patients with ASD in the Health Administration. It was verified that all cases were classified as ASD in both the healthcare and educational settings.

What instruments were used? In the majority of cases, the risk of ASD was identified using the ADI-R [55] and the ADOS-2 [56], which were employed to obtain information from parents and children, respectively. Information was obtained on three specific domains: (a) language and communication; (b) reciprocal social interactions; and (c) restricted, repetitive, or stereotyped behaviors and interests. Information on emotional or behavioral difficulties was gathered through interviews with family members and teachers. In instances where the children were under the care of other professionals, the family was required to submit their assessments. When requested by the family, interviews with these professionals were conducted.

Cognitive development was assessed with the WPPSI-IV (Wechsler Preschool and Primary Intelligence Scale) [57] when the student’s condition allowed. In instances where this was not possible, the Child Psychomotor Development Scale (Brunet–Lezine Scale) [58] was employed in conjunction with the Battelle Development Scale [59,60]. Additionally, other instruments were utilized to corroborate the findings of the initial assessments or to examine specific elements of interest pertinent to each individual case.

### 2.3. Variables and Analysis Procedures

The following sociodemographic variables were used: (1) sex (boy, girl) and (2) age (three, four, and five years). As independent variables, (3) cognitive development, (4) psychomotor development, (5) linguistic communicative development, and (6) social and emotional development were used. The schooling modality (0: ordinary or 1: specific) was considered a dependent variable. For the cognitive development variable (3), four categories were considered (Figure 5):(3.a) Cognitive development is significantly lower compared to their age. Needs constant care;(3.b) Lower cognitive development compared to their age. Needs intervention related to one of the cognitive processes (perception, attention, memory, thinking, and intelligence);(3.c) Average or superior cognitive development with respect to their age. Needs specific intervention in cognitive processes (perception, attention, memory, thinking, and intelligence;(3.d) Does not need specific attention in relation to cognitive development.

For the variable psychomotor development (4), four categories were considered (Figure 6):5.(4.a) Severe Psychomotor delay. Psychomotor development is less than expected for their chronological age;6.(4.b) Simple psychomotor delay. Psychomotor development is less than expected for their chronological age;7.(4.c) Specific psychomotor difficulties. Specific psychomotor delay: running, jumping, drawing, no deficit;8.(4.d) Functional psychomotor development. It does not need specific attention.

The Communicative-linguistic development was structured into (5.1) intentional communication, (5.2) expressive language, and (5.3) comprehensive language. For the intentional communication (5.1), the following six categories were proposed (Figure 7):9.(5.1.a) Does not show intentional communication; no communicative interaction. Does not initiate useful communicative interactions. Needs intense specific attention;10.(5.1.b) With communicative intent but lacks strategies for effective interaction. The communicative intention is appreciated, although there are no strategies to initiate an effective interaction;11.(5.1.c) Develops request functions through instrumental acts, signaling, use of learned gestures, or signs or oral productions. Needs specific attention;12.(5.1.d) Develop declarative acts through signs, gestures, or oral productions to share information. Develop declarative acts through signaling, handing or showing objects, use of learned gestures or signs, and/or some oral productions aimed at sharing information about focuses of interest or attention;13.(5.1.e) Develop comment categories in oral formats. Has structural difficulties. Develops categories of commentary in oriented oral formats, although structural difficulties limit the use of language as a means of social interaction. Needs specific attention;14.(5.1.f) Intentional communication is functional. Does not need specific attention.

For expressive language (5.2), six categories were under consideration (Figure 7):15.(5.2.a) He has no expressive repertoire of referential words; in some cases, he can make oral approximations. Needs intensive specific attention. He makes some oral approximations but does not present an expressive repertoire of referential words. Needs intense specific attention;16.(5.2.b) Limitations in speech motor control. Use of Augmentative–Alternative Communication Systems (AAC) or technical aids. Presents limitations in the motor control of speech that make it necessary to use augmentative or alternative communication systems and/or technical aids. Needs specific attention;17.(5.2.c) Difficulties in using phonological rules. Unintelligible expression. He uses words or oral productions, although his difficulties regarding the integration of the phonological rule system cause an unintelligible expression if contextual support is lacking. Needs specific attention;18.(5.2.d) Syntactic reduction. The disorganization of the statement limits the representation of the language. He uses simple sentences, but the syntactic reduction and disorganization of the statements limit the representation capacity of the language. Needs specific attention;19.(5.2.e) Morphological errors. The lack of cohesion of the statements limits the discursive organization and information of reality that is not present. Uses elaborate phrases, although errors in morphological use and cohesion of statements limit the organization of the discourse and the ability to report on non-present reality. Needs specialized care;20.(5.2.f) Functional expressive language. Does not need specific attention.

With regard to comprehensive language (5.3), six categories were considered (Figure 7):21.(5.3.a) Interpretation of visual/contextual cues. Lack of understanding of the oral code. His understanding is linked to the interpretation of visual or contextual cues; therefore, there is no understanding of the oral code. Need specialized care;22.(5.3.b) Understands simple commands and identifies objects, actions, places, and people in the immediate environment. Needs specific attention;23.(5.3.c) Understands orders and sequences of instructions that allow him to access the academic routines of the classroom and interprets short, directed stories that are supported with graphic material. Needs specific attention;24.(5.3.d) Understands simple stories, although shows difficulties when the complexity increases or the information is presented only using oral support (without visual support). Needs specific attention;25.(5.3.e) Understands texts and stories, although shows difficulty in interpreting inferential information about intentions, beliefs, or emotions. Needs specific attention;26.(5.3.f) Comprehensive language development is functional. Does not need specific attention.

For social and emotional development (6), five categories were considered (Figure 8):27.(6.a) Presents persistent and serious emotional and behavioral difficulties (physical and/or verbal violence against people and/or objects, problems respecting social norms, and other disruptive behaviors that hinder positive social interaction) that require interventions in crisis situations. Needs specific attention and permanent supervision;28.(6.b) Presents emotional and behavioral difficulties (externalizing or inhibition) that interfere with the teaching–learning and/or coexistence process. Needs continued specific attention;29.(6.c) Presents difficulties in initiating and/or maintaining social relationships with peers or adults in the context of the classroom and recreational spaces. Needs specific continuous or punctual attention;30.(6.d) Presents emotional difficulties related to low expectations, self-concept, and belief in one’s own abilities, which interferes with the teaching–learning process. Needs specific continuous or punctual attention.31.(6.e) Functional social and emotional development. Does not require specific attention.

The anonymous psychopedagogical evaluation reports were encoded by two reviewers using the content analysis application ATLAS.ti v.7 [61]. In the event of a discrepancy, the two reviewers jointly reviewed the material and reached a consensus. The encoding results were then transferred to a data table in SPSS v.28. The statistical analysis was conducted using artificial neural networks and classification trees. Briguglio et al. [62] demonstrated the utility of machine learning techniques (classification trees and artificial neural networks) in predicting autism spectrum disorders from the Autism Diagnostic Observation Schedule, second edition (ADOS-2), scores, and retrospective data. In the present study, we aim to elucidate the role of the psycho-evolutionary characteristics of children with ASD in the decision-making process regarding their early childhood education modality (ordinary or specific).

### 2.4. Analysis of Artificial Neural Networks

The objective was to ascertain which independent variables were of the greatest importance in the estimation of the modality of schooling. To this end, an analysis of neural networks was conducted [63] using 70% of the sample for training (268) and 30% of the test (115) [SPSS > Analyze > Neural Networks > Multilayer perceptron].

In other words, the network was initially trained with 70% of the data, and the parameters of the neural network were adjusted in order to optimize the weights and biases, with the objective of minimizing the prediction error. The 30% test data allowed us to adjust the hyperparameters (hidden layer with seven neurons), detect model overfitting, and determine the optimal point at which to stop training [64]. An automatic selection of the architecture was chosen, which allowed the computational algorithm to select the procedures (activation functions) that generated a simpler model (greater parsimony).

From the observed variables (input layer), a hidden layer was generated using a hyperbolic tangent activation function [tanh(x) = (2/(1 + e^−2x^)) − 1] [65], with output values for the hidden layer between 1 and −1 (which involves the normalized adjustment of the values of the observed variables). The values of the hidden layer were activated with a softmax function [softmax(x) = ln(1 + *e^x^*)] for the two categories of the dependent variable (schooling modality) (Figure 9).

The initial stage of the process entailed the identification of the most significant independent variables, followed by the normalization of the regression coefficients to percentages. The receiver operating characteristic (ROC) curve chart was constructed to evaluate the suitability of the model in the relationship between sensitivity [true positive /(true positives + false negative)] and specificity (true negative/(true negative + false positive)] across the area under the curves corresponding to the two categories of the dependent variable [66].

### 2.5. Classification Tree Analysis

An association analysis between variables was conducted using the classification tree technique with chi-squared as the measure. The decision method employed was Fisher’s exact test. The association between pairs of variables was analyzed to ascertain whether the proportions of one variable differed depending on the value of the other variable.

The dependent variable was the schooling modality (ordinary or specific), while the independent variables were those that presented the greatest weight in the analysis of neural networks. Classification tree analysis [SPSS > Analyze > Classification > Tree) was carried out using a comprehensive growth method EXHAUSTIVE CHAID [67]: 70% of the sample was used as training and 30% as proof [68]. As a significant value for the Pearson chi-square statistic, the value of 0.05 (95% CI) was taken [69]. The procedure commences with the global or parent sample, which is treated as a single group. This is then divided into subgroups, which are characterized by a greater intergroup variance and a lower intragroup variance.

## 3. Results

In examining the relationship between schooling modality and cognitive development, it was found that of the 381 Psychopedagogical Evaluation Reports, 172 corresponded to specific modalities (45.1%), while 209 were associated with ordinary modalities (54.9%) (Table 1).

A total of 128 boys and 46 girls aged between three and five years old were enrolled in specific modalities. This equates to 42.9% of boys and 54.12% of girls with autism spectrum disorder in this age group being enrolled in specific modalities. At the age of three, 42% of boys (34) and 50% of girls (12) enter school for the first time in the second cycle of early childhood education in specific modalities.

With regard to the cognitive development variable, it was established that 92.5% of the subjects exhibiting markedly inferior cognitive development relative to their age group were enrolled in specific modalities (Table 2). The ten subjects with significantly low cognitive levels who are educated in ordinary modalities are distributed as follows: five are boys, and five are girls. The group comprises six three-year-olds, one four-year-old, and three five-year-olds. The five children who have been identified as having specific learning needs and a medium or high level of cognitive development are placed in special units that are specifically designed to cater to children with autism spectrum disorders. Of the five children, three are five years old, and the other two are four years old.

With regard to psychomotor development, 75.2% of the subjects exhibited functional psychomotor development. Only seven of the 174 subjects who received training in specific modalities were classified as having severe psychomotor retardation (4.02%) (Table 3).

Concerning intentional communication, it was found that the vast majority of students with ASD (98.16%) exhibited some degree of difficulty. Of these, 129 (33.86%) demonstrated a lack of communicative intention, while 131 (34.38%) exhibited communicative abilities but lacked the requisite strategies for effective interaction. Consequently, it can be concluded that 68.24% of children with ASD between the ages of 3 and 5 exhibited significant communication difficulties (Table 4).

It is common for individuals within this subject group to have trouble with expressive language. Of the 222 subjects included in this study, 58.27% demonstrated an absence of an expressive repertoire or were limited to unintelligible oral approximations (Table 5).

In terms of comprehensive language, the findings align with those observed in the other two dimensions of language, namely, communicative intention and expressive language. A total of 123 subjects (32.28%) demonstrated an inability to comprehend oral language, while 161 subjects (42.26%) exhibited comprehension of only simple commands. This indicates that 74.54% of the subjects exhibited significant difficulties in oral comprehension (Table 6).

Respecting social and emotional development, students who experience severe and persistent difficulties (including physical and/or verbal violence against persons and/or objects, as well as very severe and persistent disruptive behavior) and who require specialized intervention in crisis situations are most often educated in specific modalities (Table 7).

For analyzing neural networks, the training sample comprised 259 cases (68%), while the testing sample consisted of 122 cases. The Neural Network Analysis indicated that the most relevant variables in determining the school modality were cognitive development (323; 100%) and comprehensive language (176; 54.6%) (Table 8).

The ROC (Receiver Operating Characteristic) curve offered a sensitivity/specificity ratio (area under the curve) for the two categories of the variable schooling mode of 0.896, indicating a strong predictive model. It is possible to assume the explanatory character of cognitive development and the comprehensive language of the variable schooling modality (Figure 10).

The classification using Classification Trees (Pearson chi-square) and EXHAUSTIVE CHAID employed as a growth method, generated a predictive model that explains the regular schooling modality at 95.2%, with a total percentage for both modalities of 83.5%. Two clusters were identified for the cognitive development variable. The significantly low level corresponded to schooling in specific modes, while the remaining 21% were found in specific modalities and 54.9% in ordinary modes. For the classification of this second group [(2) or (3) or (4)], the level of comprehensive language was relevant. The group with the lowest level of comprehension [(1) or (2)] attended school in 27.8% of specific modalities, while the group with the highest level of comprehension attended school mostly in ordinary modalities (90.4%) (Table 9).

## 4. Discussion and Conclusions

The proportion of children aged 3 to 5 years with ASD who attend school in specific modalities is considerable (45.1%). This is particularly noteworthy in the 3-year-old cohort. For the 0- to 2-year-old group, no distinct schooling modalities are envisaged; rather, the ordinary one is to be provided, irrespective of any special educational needs.

At the age of three, when children commence their formal education, these uncertainties are particularly salient. At this age, 41.3% of boys and 50% of girls with autism spectrum disorder are enrolled in specific school modalities. In general, 42.9% of boys and 54.12% of girls aged 3 to 5 years with ASD were enrolled in special education.

In terms of cognitive development, it was observed that the individuals with the most impaired cognitive abilities were predominantly affected in specific modalities (92.2%). Additionally, the five children with a normal or superior level of cognitive development who were educated in specific modalities were placed in specialized units for ASD. This is a significant consideration, given that although the reference is that the student who can benefit from ordinary modalities should not be educated in specific modalities, there may be instances where families and professionals may deem it beneficial for the student to receive education in specific units. This situation was observed in five of the 57 cases that were classified as having a medium or high level of cognitive development, representing 8.77% of the total cases.

Regarding psychomotor development, it was observed that only 4.02% of those who received education in specific modalities (n = 7) exhibited severe psychomotor retardation. However, difficulties related to communicative intentionality were observed in all cases (98.16%). It was found that of the students enrolled in specific modalities (172), 89 had no communicative intention, and 59 had significant difficulties. This indicates that 86.04% of those who received education in specific modes exhibited notable difficulties in communication.

With a view to expressive language, it was found that 78% of children in specific modalities exhibited either no expressive repertoire or a restricted range of oral approximations. However, 39.6% of those lacking an expressive repertoire were in ordinary modalities. In terms of comprehensive language, those who are unable to comprehend oral language or whose understanding is contingent upon contextual cues are primarily enrolled in specific modalities (75.6%).

Among children who demonstrate comprehension of basic instructions, 58.4% are enrolled in conventional modalities, while 41.6% are enrolled in specific modalities. The remaining children (who exhibit superior comprehensive language abilities) are enrolled in general conventional modalities. Although it was expected that children with lower levels of communicative intention, no expressive referential language, and lower levels of comprehensive language would be taught in specific modalities (76.8%), 19 (23.2%) were found to be in ordinary modalities.

A total of 30 cases were identified that exhibited a markedly low profile of communicative intentionality, a notably low level of expressive language, and a comprehensive language that enabled them to comprehend simple commands. In this case, 17 (56.7%) subjects had received education in specific modalities, while 13 (43.3%) had received education in ordinary modes.

In terms of social and affective development, those who presented a profile of persistent and severe emotional and behavioral difficulties (including physical and/or verbal violence against people and other serious disruptive behaviors) (n = 35) were primarily enrolled in specific modalities (n = 29; 82.9%), with a minority enrolled in ordinary modalities (n = 6; 17.1%).

To ascertain which developmental variables are pertinent in elucidating the schooling mode, the neural network analysis identified cognitive development and comprehensive language as especially significant. In other words, these two variables explain 83.5% of the mode of schooling in general and 95.2% of the mode of schooling in particular (Figure 11).

The classification tree using Pearson’s Chi-squared as the classification statistic showed that the category of having a significantly impaired level of cognitive development was the most accurate predictor of schooling in specific modalities (n = 119; 92.2%).

For the other levels of cognitive development, the most relevant criterion for the feasibility of schooling in specific modalities (27.8%) or ordinary (72.2%) was the absence of comprehension of the oral code or the limited comprehensibility of language, which was restricted to the understanding of simple commands. Cognitive development is above the lowest level, and with a greater comprehensive language, the modalities of ordinary schooling are generalized (90.4% in the ordinary modalities and 9.6% in the specific modalities) (Figure 12).

The principal limitation of this work is the geographical and administrative context. The schooling decisions may be different in other Autonomous Communities of Spain, and perhaps even in other provinces of the same Autonomous Community (Andalusia). Further research could investigate the consistency of educational choices across other Andalusian provinces and other Spanish autonomous communities. Additionally, the role of cognitive development and language comprehension in the schooling of children aged 3–5 could be examined. It is also pertinent to consider whether there are differences in the schooling of children with ASD as they progress through their education. That is, if the categorization of variables related to the personal development of children with ASD, the transition to primary education, and later to secondary education results in significant changes in the type of schooling.

Finally, this work provides relevant information for families facing schooling of 3-year-old children with ASD, as well as for education professionals and administrators, who can reflect on their preschooling practices for these children and make decisions regarding the resources to be provided for the coming years. This paper serves as a call to action, advocating for the implementation of personalized education plans developed by specialist teachers and the creation of tailored training programs for practicing educators. These initiatives aim to establish individualized spaces and resources.

## Figures and Tables

**Figure 1 brainsci-14-01167-f001:**
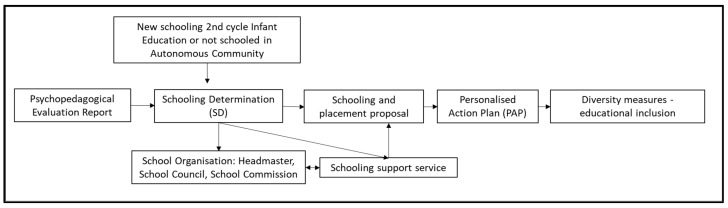
Procedure for the schooling of pupils with SEN in Almería (own elaboration).

**Figure 2 brainsci-14-01167-f002:**
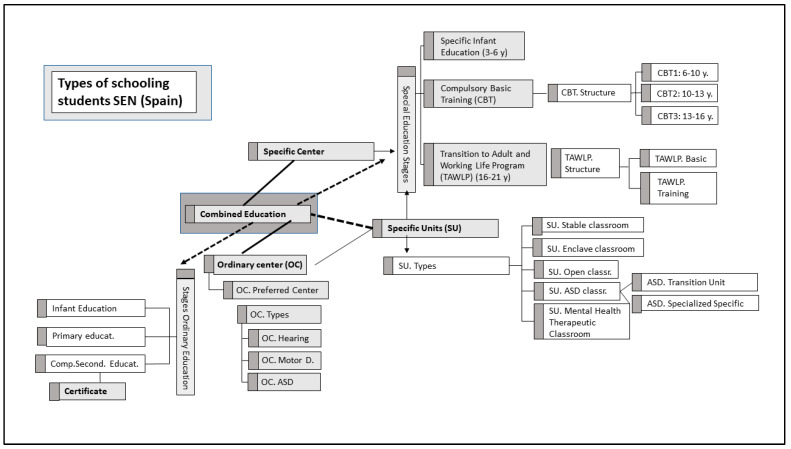
Types of schooling students SEN in Almería (own elaboration).

**Figure 3 brainsci-14-01167-f003:**
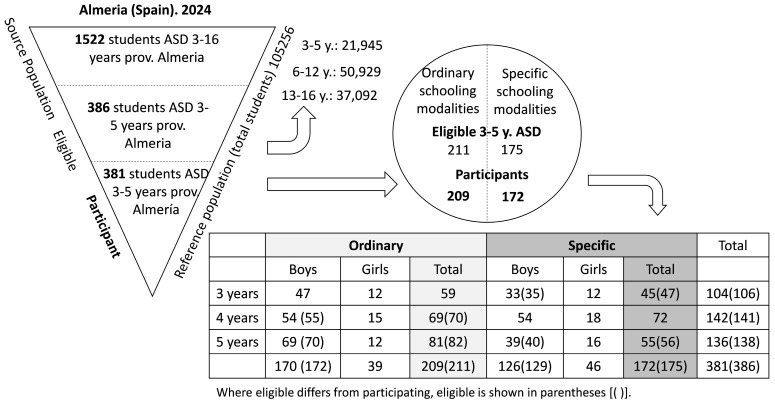
Source population, participants by sex, age, and schooling modalities (own elaboration).

**Figure 4 brainsci-14-01167-f004:**
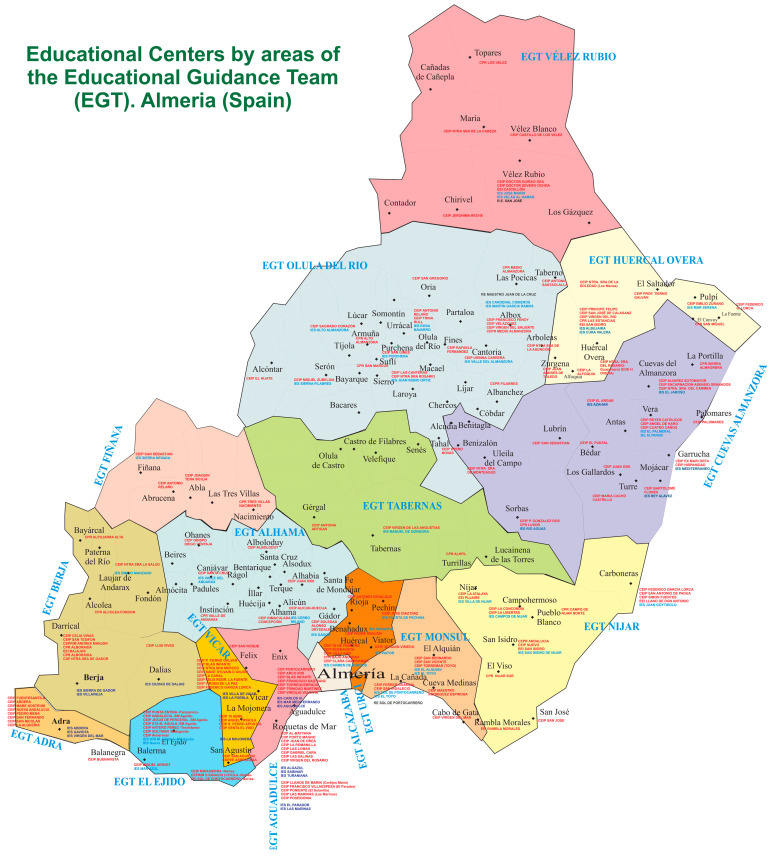
Educational Centers by areas of each Educational Guidance Team (EGT), Almería, Spain (own elaboration).

**Figure 5 brainsci-14-01167-f005:**
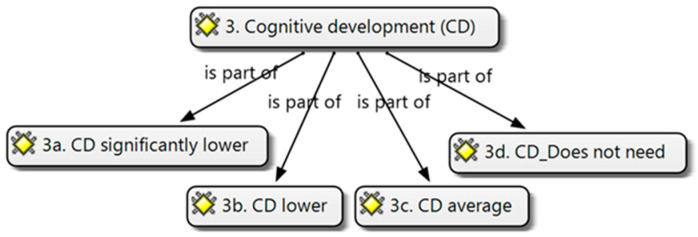
Cognitive development categories (own elaboration).

**Figure 6 brainsci-14-01167-f006:**
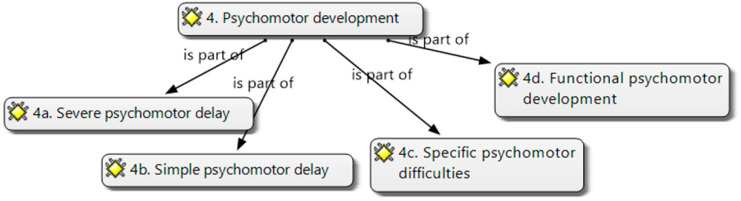
Psychomotor development categories (own elaboration).

**Figure 7 brainsci-14-01167-f007:**
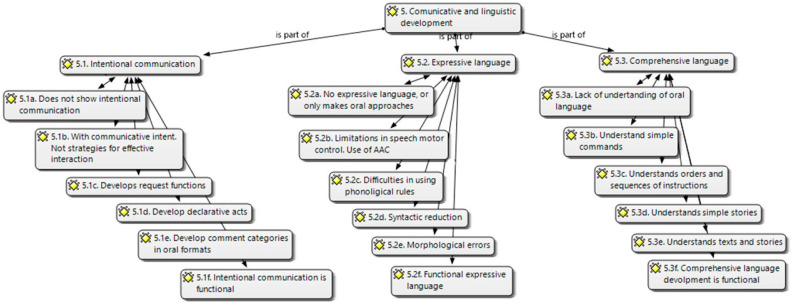
Communicative and linguistic development categories (own elaboration).

**Figure 8 brainsci-14-01167-f008:**
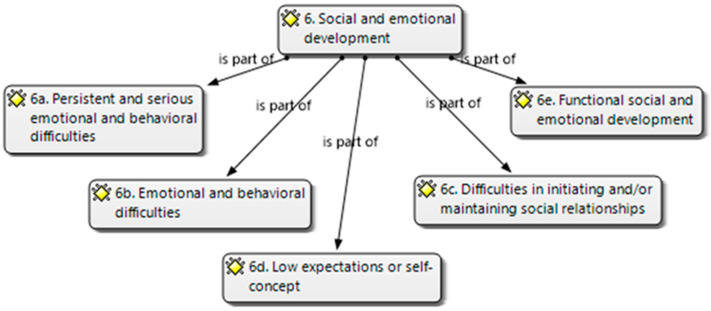
Social and emotional development categories (own elaboration).

**Figure 9 brainsci-14-01167-f009:**
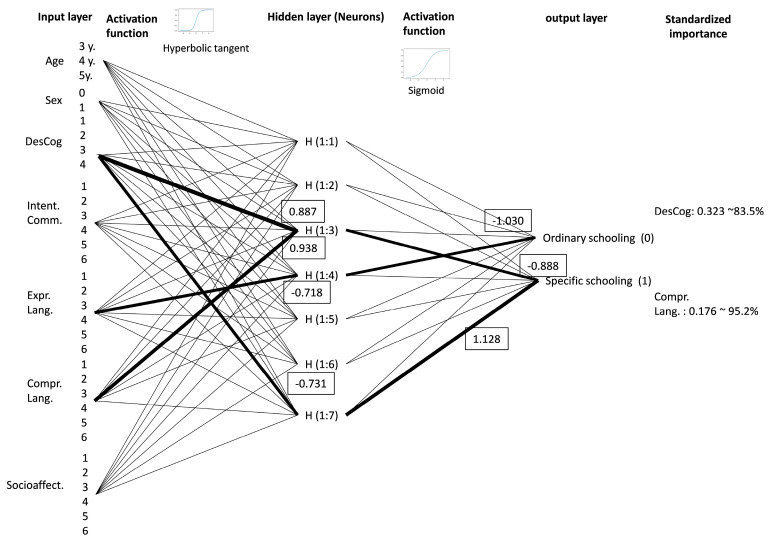
Structure of artificial neural network analysis (own elaboration).

**Figure 10 brainsci-14-01167-f010:**
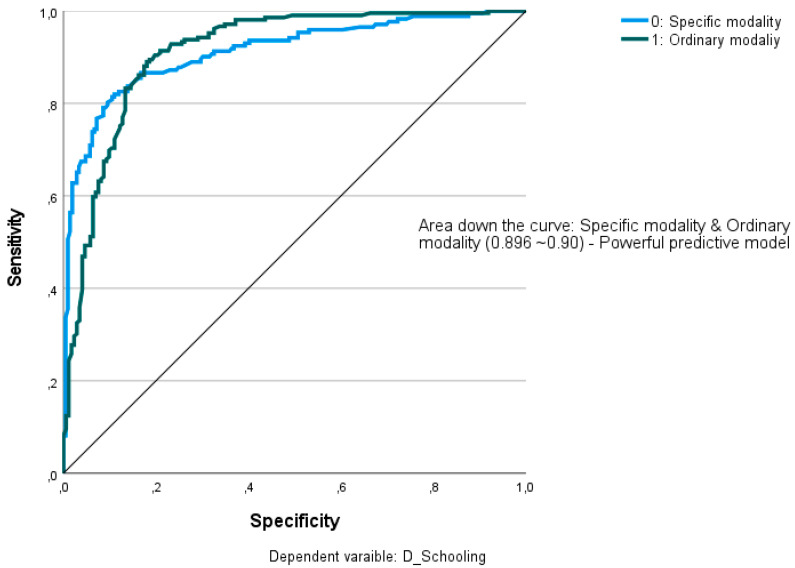
ROC curve for the predictive model of the Schooling Modality variable, by the Cognitive Development and Comprehensive Language variables.

**Figure 11 brainsci-14-01167-f011:**
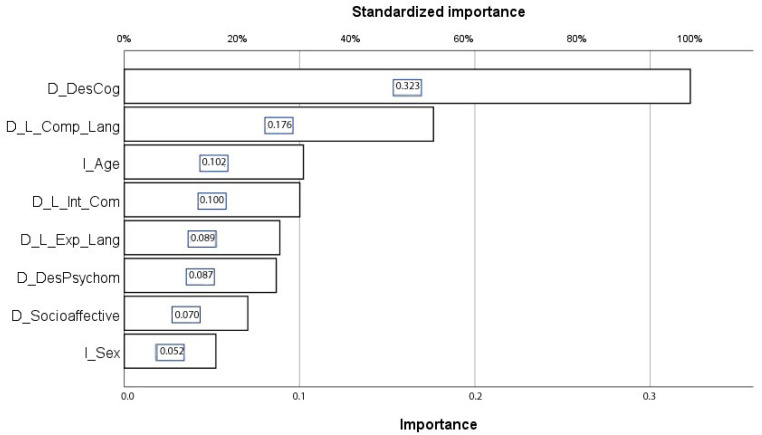
Importance of personal development variables in children with ASD in determining the type of schooling (3–5-year-old group). D_DesCog: Cognitive development; D_L_Comp_Lang: Comprehensive language; I_Age: Age; D_L_Int_Com: Communicative intention; D_L_E.

**Figure 12 brainsci-14-01167-f012:**
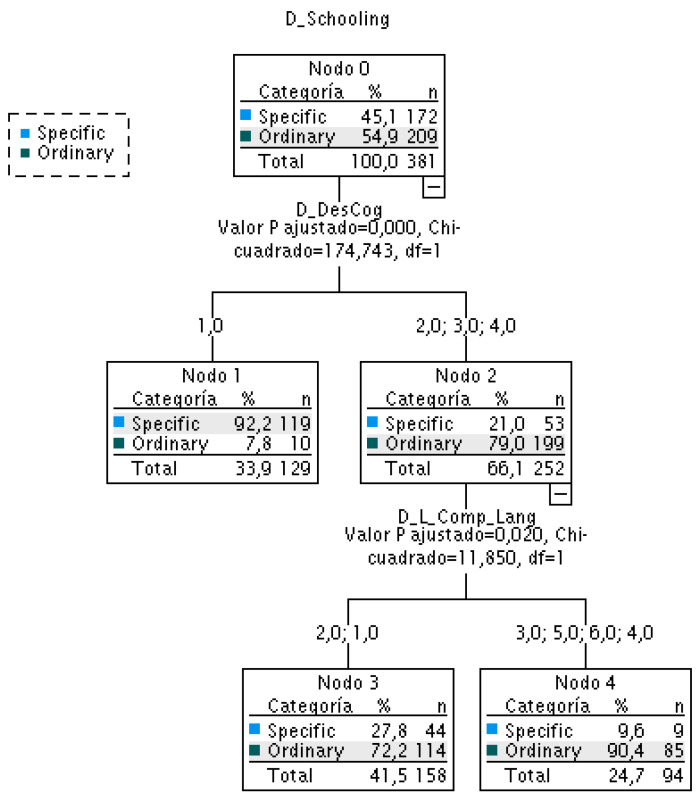
Classification tree of schooling modalities according to the variables of cognitive development and comprehensive language.

**Table 1 brainsci-14-01167-t001:** Sex, Age, and Schooling Modality.

Sex	Age	Schooling Modality		
		Specific	Ordinary	
Boys	3 years	33 (41.3%)	47 (58.8%)	80
	4 years	54 (50.0%)	54 (50.0%)	108
	5 years	39 (36.1%)	69 (63.9%)	109
Girls	3 years	12 (50.0%)	12 (50.0%)	24
	4 years	18 (54.5%)	15 (45.5%)	33
	5 years	16 (57.1%)	12 (42.9%)	28
		172 (45.1%)	209 (54.9%)	381

**Table 2 brainsci-14-01167-t002:** Cognitive development and schooling modality.

Cognitive Development	Schooling Modality		Total
	Specific	Ordinary	
(1) Significantly lower	119 (92.2%)	10 (7.8%)	129
(2) Lower	48 (24.6%)	147 (75.4%)	195
(3) Medium	4 (8.9%)	41 (91.1%)	45
(4) Does not need	1 (8.3%)	11 (91.7%)	12
	174	210	381

**Table 3 brainsci-14-01167-t003:** Psychomotor development and schooling modality.

Psychomotor Development	Schooling Modality		Total
	Specific	Ordinary	
(1) Severe psychomotor delay	7 (46.7%)	8 (53.3%)	15
(2) Simple psychomotor delay	41 (70.7%)	17 (29.3%)	58
(3) Specific psychomotor difficulties	11 (50.0%)	11 (50.0%)	22
(4) Functional psychomotor development	113 (39.5%)	209 (54.9%)	286
	172	210	381

**Table 4 brainsci-14-01167-t004:** Intentional communication and schooling modality.

Psychomotor Development	Schooling Modality		Total
	Specific	Ordinary	
(1) Does not show intentional communication	89 (69.0%)	40 (31.0%)	129
(2) With communicative intent but lacks strategies for effective interaction	59 (45.0%)	72 (55.0%)	131
(3) Develops request functions	17 (27.0%)	46 (73.0%)	63
(4) Develop declarative acts	5 (15.6%)	27 (84.4%)	32
(5) Develop comment categories in oral formats	2 (10.5%)	17 (89.5%)	19
(6) Intentional communication is functional	0 (0.0%)	7 (100%)	7
	172	209	381

**Table 5 brainsci-14-01167-t005:** Expressive language and schooling modality.

Expressive Language	Schooling Modality		Total
	Specific	Ordinary	
(1) Without expressive repertoire	134 (60.4%)	88 (39.6%)	222
(2) Limitations in speech motor control	16 (69.6%)	7 (30.4%)	23
(3) Phonological difficulties	11 (17.5%)	52 (82.5%)	63
(4) Syntactic difficulties	10 (19.2%)	42 (80.8%)	52
(5) Morphological difficulties	1 (7.1%)	13 (92.9%)	14
(6) Functional expressive language	0 (0.0%)	7 (100%)	7
	172	209	381

**Table 6 brainsci-14-01167-t006:** Comprehensive language and schooling modality.

Comprehensive Language	Schooling Modality		Total
	Specific	Ordinary	
(1) Lack of understanding of the oral code.	93 (75.6%)	30 (24.4%)	123
(2) Understands simple commands.	67 (41.6%)	94 (58.4%)	161
(3) Understands simple stories. Needs visual aids.	9 (15.3%)	50 (84.7%)	59
(4) Understands simple stories. Does not require visual aids.	1 (7.7%)	12 (92.3%)	13
(5) Includes texts and stories. Difficulties in inferring meanings.	2 (10.0%)	18 (90.0%)	20
(6) Comprehensive functional language.	0 (0.0%)	5 (100%)	5
	172	209	381

**Table 7 brainsci-14-01167-t007:** Social and emotional development and schooling modality.

Social and Emotional Development	Schooling Modality		Total
	Specific	Ordinary	
(1) Persistent and severe emotional and behavioral difficulties	29 (82.9%)	6 (17.1%)	35
(2) Emotional and behavioral difficulties that interfere with learning and/or coexistence	56 (53.8%)	48 (46.2%)	104
(3) Difficulties in initiating and/or maintaining social relationships with peers or adults in the educational center	84 (37.0%)	143 (63.0%)	227
(4) Emotional difficulties due to low expectations or low self-concept	2 (22.2%)	7 (77.8%)	9
(5) Functional social and affective development	1 (16.7%)	5 (83.3%)	6
	172	209	381

**Table 8 brainsci-14-01167-t008:** Importance of independent variables (Neural Network Analysis).

Independent Variable	Coefficient of Importance	Normalized Importance (%)
Age	0.102	31.6%
Sex	0.052	16.2%
Cognitive development	0.323	100%
Psychomotor development	0.087	26.8%
Intentional communication	0.100	31.0%
Expressive language	0.089	27.5%
Comprehensive language	0.176	54.6%
Social and emotional development	0.070	21.8%

**Table 9 brainsci-14-01167-t009:** Chi-square values for the division categories that explain the distribution of cases of the schooling modality variable, based on the variables Cognitive Development and Comprehensive Language.

Node	Specific N (%)	Ordinary N (%)	Predicted Category	Variable	Sig.	Chi-Square	Division Values
0	172 (45.1%)	209 (54.9%)	Ordinary				
1	119 (92.2%)	10 (7.8%)	Specific	Des. cognt	<0.001	174.74	(1)
2	53 (21.0%)	199 (79.0%)	Ordinary	Des. cognt	<0.001	174.74	(2) or (3) or (4)
3	44 (27.8%)	114 (72.2%)	Ordinary	Compr. Lang	0.020	11.85	(1) or (2)
4	9 (9.6%)	85 (90.4%)	Ordinary	Compr. Lang	0.020	11.85	(3) or (4) or (5)

## Data Availability

The raw data supporting the conclusions of this article will be made available by the authors upon request.

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
