# Peer review of "Relevant Factors in the Schooling of Children with Autism Spectrum Disorder in Early Childhood Education"

_brainsci, 2024, doi:10.3390/brainsci14121167_

Round 1
Reviewer 1 Report
Comments and Suggestions for Authors
The manuscript presented by the authors addresses a significant issue in autistic students education. While the content is intriguing and relevant, the overall structure of the manuscript presents challenges in terms of readability and coherence. The predominant use of short sentences, often spanning only one or a few lines, disrupts the flow of the text, making it difficult for the reader to follow the logical progression of ideas. In some instances, these short sentences appear disjointed and lack clear connections between them, further contributing to the lack of coherence.
Although the topic holds promise and has potential relevance to the field, this reviewer finds that the current manuscript is not yet suitable for publication in Brain Sciences. A more substantial revision is necessary to enhance the manuscript’s organization and clarity. My primary recommendation is that the authors undertake a thorough rewrite of the manuscript, focusing on improving its structure and ensuring a more fluid and logical arrangement of ideas.
One particular issue that requires attention is the use of quotation marks to indicate that a portion of text is directly taken from another source, as seen at the beginning of the Introduction section. This practice should be avoided, and the text should instead be paraphrased and appropriately referenced.
Comments on the Quality of English LanguageA detailed review of the grammar is essential to improve the overall readability of the manuscript.
Author Response
The manuscript presented by the authors addresses a significant issue in autistic students education. While the content is intriguing and relevant, the overall structure of the manuscript presents challenges in terms of readability and coherence. The predominant use of short sentences, often spanning only one or a few lines, disrupts the flow of the text, making it difficult for the reader to follow the logical progression of ideas. In some instances, these short sentences appear disjointed and lack clear connections between them, further contributing to the lack of coherence.
Although the topic holds promise and has potential relevance to the field, this reviewer finds that the current manuscript is not yet suitable for publication in Brain Sciences. A more substantial revision is necessary to enhance the manuscript’s organization and clarity. My primary recommendation is that the authors undertake a thorough rewrite of the manuscript, focusing on improving its structure and ensuring a more fluid and logical arrangement of ideas.
Authors reply: The paper has been revised to improve the organization and clarity by reducing the use of short sentences and making the structure of the manuscript more fluid.
One particular issue that requires attention is the use of quotation marks to indicate that a portion of text is directly taken from another source, as seen at the beginning of the Introduction section. This practice should be avoided, and the text should instead be paraphrased and appropriately referenced.
Authors reply: Author's reply: Only direct quotations that give voice to the families and consist of a transcription of their statements have been kept. In this case, the direct quotations have been modified in order to adapt them to the standards of presentation (indentation in the case of direct quotations longer than 40 words). The rest of the quotations have been paraphrased, as suggested by the reviewer.

Reviewer 2 Report
Comments and Suggestions for Authors
The topic of this manuscript seems to be to examine the influence of two schooling modalities, namely the specific and the ordinary, on the educational outcomes of children with ASD aged 3-5. This manuscript is meaningful for the authors and their team have paid a great effort on practical work. However, the existing manuscript presents several issues that require attention.:
1. in the introduction section, there lacks sufficient background information on existing studies on the effects of different schooling modalities on children aged 3-5, particularly those with ASD. It is challenging for readers to fully comprehend the authors' purpose in conducting this research.
2. in the methodology section, despite the regional focus of the study, the distinction between the specific and the ordinary schooling modalities is not adequately delineated, and there are no discernible observational and evaluative indicators to differentiate between the two modalities. It is challenging to proceed with subsequent discussions and in-depth analyses.
3. in the data section, the total number of research samples is 381, but the total number of research samples in Table 2 is 384, while the total number of research samples in Table 3 is 382. This discrepancy must be clarified.
4. in the discussion section, there lacks further investigation into the potential reasons and evidence supporting the study's findings.
Comments on the Quality of English LanguageThe manuscript writing style and presentation form need to be optimised.
Author Response
The topic of this manuscript seems to be to examine the influence of two schooling modalities, namely the specific and the ordinary, on the educational outcomes of children with ASD aged 3-5. This manuscript is meaningful for the authors and their team have paid a great effort on practical work. However, the existing manuscript presents several issues that require attention.:
Authors reply: Thank you very much for considering this manuscript to be meaningful.
- in the introduction section, there lacks sufficient background information on existing studies on the effects of different schooling modalities on children aged 3-5, particularly those with ASD. It is challenging for readers to fully comprehend the authors' purpose in conducting this research.
Authors reply: Following this recommendation, an attempt has been made to justify the relevance of the work in more detail at the beginning of the introductory section, and a specific section has been included to highlight the purpose of the research.
- in the methodology section, despite the regional focus of the study, the distinction between the specific and the ordinary schooling modalities is not adequately delineated, and there are no discernible observational and evaluative indicators to differentiate between the two modalities. It is challenging to proceed with subsequent discussions and in-depth analyses.
Authors reply: Information has been added so that the reader can be aware of the difference between specific and ordinary schooling modalities.
- in the data section, the total number of research samples is 381, but the total number of research samples in Table 2 is 384, while the total number of research samples in Table 3 is 382. This discrepancy must be clarified.
Authors reply: The number of participants is 381 students. The discrepancy mentioned by the reviewer was not found. See Tables 1, 2 and 3.
|
Table 1. Sex * Age, and schooling modality |
||||
Sex |
Age |
Schooling modality |
|
|
|
|
|
Specific |
Ordinary |
|
|
Boys |
3 years |
33 (41.3%) |
47 (58.8%) |
80 |
|
|
4 years |
54 (50.0%) |
54 (50.0%) |
108 |
|
|
5 years |
39 (36.1%) |
69 (63.9%) |
109 |
|
Girls |
3 years |
12 (50.0%) |
12 (50.0%) |
24 |
|
|
4 years |
18 (54.5%) |
15 (45.5%) |
33 |
|
|
5 years |
16 (57.1%) |
12 (42.9%) |
28 |
|
|
|
172 (45.1%) |
209 (54.9%) |
381 |
|
Table 2. Cognitive development and schooling modality |
|||
|
Cognitive development |
Shooling modality |
Total |
|
|
|
Specific |
Ordinary |
|
|
(1) Significantly lower |
119 (92.2%) |
10 (7.8%) |
129 |
|
(2) Lower |
48 (24.6%) |
147 (75.4%) |
195 |
|
(3) Medium |
4 (8.9%) |
41 (91.1%) |
45 |
|
(4) Does not need |
1 (8.3%) |
11 (91.7%) |
12 |
|
|
174 |
210 |
381 |
|
Table 3. Psychomotor development and schooling modality |
|||
|
Psychomotor development |
Shooling modality |
Total |
|
|
|
Specific |
Ordinary |
|
|
(1) Severe psychomotor delay |
7 (46.7%) |
8 (53.3%) |
15 |
|
(2) Simple psychomotor delay |
41 (70.7%) |
17 (29.3%) |
58 |
|
(3) Specific psychomotor difficulties |
11 (50.0%) |
11 (50.0%) |
22 |
|
(4) Functional psychomotor development |
113 (39.5%) |
209 (54.9%) |
286 |
|
|
172 |
210 |
381 |
- in the discussion section, there lacks further investigation into the potential reasons and evidence supporting the study's findings.
Authors reply: We believe that the evidence provided following the analysis performed sufficiently supports and backs up the results.

Reviewer 3 Report
Comments and Suggestions for Authors
The research work is adequately presented, but it would be
It is advisable that the relevance of the study also considers aspects
related to the treatment required by Autism. Justify by
what Colombia was chosen. The references cited are mostly
recent publications (from the last 5 years); but about 30% of
The quotes are more than 5 years old. The results of the
manuscript coincide with what is detailed in the methods section. He
methodological approach can be improved by referring to the approach,
method, techniques and instruments used. The figures and tables
They display the data in an appropriate way that is easy to interpret and understand.
I consider that the conclusions should be improved in accordance with the
objectives proposed in the research work, to emphasize
the results obtained, the gaps and the findings.
Author Response
The research work is adequately presented, but it would be. It is advisable that the relevance of the study also considers aspects related to the treatment required by Autism. Justify by what Colombia was chosen. The references cited are mostly recent publications (from the last 5 years); but about 30% of. The quotes are more than 5 years old. The results of the manuscript coincide with what is detailed in the methods section. He methodological approach can be improved by referring to the approach, method, techniques and instruments used. The figures and tables. They display the data in an appropriate way that is easy to interpret and understand. I consider that the conclusions should be improved in accordance with the objectives proposed in the research work, to emphasize the results obtained, the gaps and the findings.
Authors reply: There must be a problem with the document you have received, since there is no reference to Colombia in our work. In any case, we have improved the discussion and conclusions section.

Reviewer 4 Report
Comments and Suggestions for Authors
To the AAs
Schooling in ASD children is certainly a major challenge. In their Ms entitled “Relevant Factors in the Schooling of Children with Autism Spectrum Disorder in Early Childhood Education” the Authors specifically address the factors in the language development, cognitive development, and socio-emotional development in pre-school students (aged 3-5) affected by ASD in Almeria, Spain. Ordinary or specific schooling were evaluated on anonymous psycho pedagogical reports . Less than half of the boys (42.9%) and just above half of the girls (54.12%) were enrolled in specific schooling modalities. Cognitive development and comprehensive language were identified by the Authors as the two variables that best explained the assignment of preschool children with ASD to specific schooling.
Although the current Ms is certainly filled with a lot of data and details and has likely some merits, this reviewer finds the Ms, overall, quite confusing.
In a constructive perspective, I will try to summarize 3 main issues evidenced:
1. Introduction is extremely lengthy, with several sub-headings and narrative sections. I encourage the Authors to consistently shorten this section where the flow of concepts is often wandering and very difficult for the reader to understand. In particular, I would suggest the Authors to reduce the section aimed at the ASD prevalence given that the study is not aimed at epidemiology. Specifically, the aims of the study should be better clarified. Please re-edit it extensively.
2. Discussion is very confusing. Please re-edit it extensively, and better clarify the relevance of the observed findings.
3. There are several grammar and style imperfections with even some interspersed spanish words, which are in need of correction.
Comments on the Quality of English LanguageThere are several grammar and style imperfections with even some interspersed spanish words, which are in need of correction.
Author Response
Schooling in ASD children is certainly a major challenge. In their Ms entitled “Relevant Factors in the Schooling of Children with Autism Spectrum Disorder in Early Childhood Education” the Authors specifically address the factors in the language development, cognitive development, and socio-emotional development in pre-school students (aged 3-5) affected by ASD in Almeria, Spain. Ordinary or specific schooling were evaluated on anonymous psycho pedagogical reports. Less than half of the boys (42.9%) and just above half of the girls (54.12%) were enrolled in specific schooling modalities. Cognitive development and comprehensive language were identified by the Authors as the two variables that best explained the assignment of preschool children with ASD to specific schooling.
Although the current Ms is certainly filled with a lot of data and details and has likely some merits, this reviewer finds the Ms, overall, quite confusing.
In a constructive perspective, I will try to summarize 3 main issues evidenced:
- Introduction is extremely lengthy, with several sub-headings and narrative sections. I encourage the Authors to consistently shorten this section where the flow of concepts is often wandering and very difficult for the reader to understand. In particular, I would suggest the Authors to reduce the section aimed at the ASD prevalence given that the study is not aimed at epidemiology. Specifically, the aims of the study should be better clarified. Please re-edit it extensively.
Authors reply: The complexity of the work requires a sufficient theoretical corpus to support the research undertaken, hence the length of the introduction. Following the reviewer's suggestions, the flow of concepts has been revised and an attempt has been made to clarify those that might be confusing to the reader. In addition, a specific heading has been added to clarify the purpose of the research.
Thus, a paragraph is included justifying the inclusion of headings related to development and difficulties of children with ASD in language, psychomotor, socio-affective and cognitive development. Similarly, the headings on schooling and schooling modalities, mental health and ASD are also justified.
- Discussion is very confusing. Please re-edit it extensively, and better clarify the relevance of the observed findings.
Authors reply: The Discussion and Conclusions section has been revised to clarify its relevance given the evidence of growth in the number of children with ASD in the context in which the research is conducted. The prevalence analysis has focused on the evolution of the population of children with ASD in the context in which the research is conducted. It is important to know (1) that its evolution has been monitored from the school year 2009 to the present, (2) that it is a group with a growth forecast of particular relevance, (3) that the increase in the identification of children with ASD in early childhood education parallels the increase in complaints from families regarding the type of schooling.
This justifies the need to improve knowledge of the profiles of children with ASD, and finally, in early childhood education, a specific schooling modality is proposed.
- There are several grammar and style imperfections with even some interspersed spanish words, which are in need of correction.
Authors reply: A professional grammar and style check has been carried out.

Reviewer 5 Report
Comments and Suggestions for Authors
Major Concerns:
-
Clarity and Focus of Research Objectives:
- The article outlines factors affecting schooling for children with Autism Spectrum Disorder (ASD) but lacks a clearly defined research question and objective. This could lead to ambiguity in interpretation. Stating a precise research question at the beginning would strengthen the article's focus.
-
Methodology Details:
- The article mentions the use of neural networks and classification trees to identify variables related to schooling decisions for children with ASD but provides limited information on the rationale behind these choices. For example, the choice of specific neural network architectures, the parameter tuning process, and why these models were preferred over others is not thoroughly explained.
- There is also insufficient information about how data from different sources were preprocessed and validated. Greater transparency about these methodological details would enhance reproducibility.
-
Discussion of Limitations:
- The article does not adequately address limitations related to the sample size and demographic scope, as it is focused on a single geographic area in Spain. A more explicit discussion of the generalizability of the findings and potential biases (e.g., selection bias) would provide a balanced perspective on the research.
-
Interpretation of Results:
- Although cognitive development and comprehensive language are identified as key predictors for schooling modality, the implications of these findings are not thoroughly explored. A deeper discussion on how these factors could inform educational policy, personalized education plans, or teacher training would be valuable for readers, especially those in education or clinical practice.
Minor Concerns:
-
Formatting and Structure:
- There are inconsistencies in formatting across sections, especially in the headings and subheadings, which disrupt the flow. Ensuring a consistent format in alignment with Brain Sciences standards will enhance readability.
- Some tables and figures lack clarity and would benefit from enhanced organization and layout, particularly those depicting categorical data for developmental assessments.
-
Terminology and Abbreviation Use:
- Certain terms (e.g., ASD, SEN) are used without clear initial definitions, which could lead to confusion for readers unfamiliar with these acronyms. Defining all abbreviations at first use and ensuring consistency throughout will improve accessibility.
-
Citation and Reference Order:
- The references in the introduction and discussion sections are not consistently ordered according to their appearance in the text. Adhering strictly to sequential citation order is essential to meet journal standards.
-
Table and Figure Legends:
- Legends are often too brief, particularly for tables summarizing developmental characteristics. Adding more descriptive captions would aid in understanding the tables and figures independently of the text.
Author Response
Major Concerns:
- Clarity and Focus of Research Objectives:
The article outlines factors affecting schooling for children with Autism Spectrum Disorder (ASD) but lacks a clearly defined research question and objective. This could lead to ambiguity in interpretation. Stating a precise research question at the beginning would strengthen the article's focus.
Authors reply: A specific section has been included to clarify the purpose of the research.
- Methodology Details:
The article mentions the use of neural networks and classification trees to identify variables related to schooling decisions for children with ASD but provides limited information on the rationale behind these choices. For example, the choice of specific neural network architectures, the parameter tuning process, and why these models were preferred over others is not thoroughly explained.
There is also insufficient information about how data from different sources were preprocessed and validated. Greater transparency about these methodological details would enhance reproducibility.
Authors reply: The usefulness of using machine learning methods (classification trees and artificial neural networks) in ASD studies was justified with examples such as the prediction of ADOS-2 scores from retrospective data (Briguglio et al., 2023), and a specific section on data processing was included.
- Discussion of Limitations:
The article does not adequately address limitations related to the sample size and demographic scope, as it is focused on a single geographic area in Spain. A more explicit discussion of the generalizability of the findings and potential biases (e.g., selection bias) would provide a balanced perspective on the research.
Authors reply: Changes are made in response to the reviewer's suggestions, with the limitations explicitly stated. We have also been more explicit in the discussion.
- Interpretation of Results:
Although cognitive development and comprehensive language are identified as key predictors for schooling modality, the implications of these findings are not thoroughly explored. A deeper discussion on how these factors could inform educational policy, personalized education plans, or teacher training would be valuable for readers, especially those in education or clinical practice.
Authors reply: This research provides relevant information on the profiles of children with ASD who attend early childhood education in ordinary modalities and those who attend special modalities.
It is very important for families to know that their children with ASD who are enrolled in early childhood education and who have a moderate or high cognitive level and a moderate or high level of language comprehension are very likely to be enrolled in ordinary modalities.
The fact that they have higher or lower levels of sensory reactivity and the resulting unexpected behavioral responses, that their level of expressive language or communicative intentionality is low, does not mean that these are the most relevant criteria in the schooling proposal.
Educational administrators, educational centers, teachers, counselors, we know that it is necessary to provide the necessary resources to ensure the well-being of children with ASD who attend regular early childhood education groups.
This requires the provision of specific resources to provide the individualized support that these students need. In particular, those who have significant difficulties in communicative intentionality, in their interactions with other classmates and with their teachers, or simply with the sensory saturation that can be generated by the usual stimuli in the classroom.
We have added some suggestions in the conclusions section.
Minor Concerns:
- Formatting and Structure:
There are inconsistencies in formatting across sections, especially in the headings and subheadings, which disrupt the flow. Ensuring a consistent format in alignment with Brain Sciences standards will enhance readability.
Some tables and figures lack clarity and would benefit from enhanced organization and layout, particularly those depicting categorical data for developmental assessments.
Authors reply: The format has been revised. Tables and figures have also been revised.
- Terminology and Abbreviation Use:
Certain terms (e.g., ASD, SEN) are used without clear initial definitions, which could lead to confusion for readers unfamiliar with these acronyms. Defining all abbreviations at first use and ensuring consistency throughout will improve accessibility.
Authors reply: All abbreviations are defined at the time of their first use.
- Citation and Reference Order:
The references in the introduction and discussion sections are not consistently ordered according to their appearance in the text. Adhering strictly to sequential citation order is essential to meet journal standards.
Authors reply: Fixed.
- Table and Figure Legends:
Legends are often too brief, particularly for tables summarizing developmental characteristics. Adding more descriptive captions would aid in understanding the tables and figures independently of the text.
Authors reply: Addition of more descriptive captions.

Round 2
Reviewer 1 Report
Comments and Suggestions for Authors
Authors responded appropriately to all my comments, I do not have further suggestions.